 SHORT REPORT

# A new pipeline SPICE identifies novel JUN-IKZF1 composite elements

**Peng Li[1]\*[†], Sree Pulugulla[2][†], Sonali Das[2], Jangsuk Oh[2], Rosanne Spolski[2], Jian-Xin Lin[2], Warren J Leonard[2]\***

[1]Amgen Inc, Rockville, United States; [2]Laboratory of Molecular Immunology, National Heart, Lung, and Blood Institute, NIH, Bethesda, United States

## eLife assessment

This **valuable** study presents a screening pipeline (SPICE) for detecting DNA motif spacing preferences between TF partners. SPICE predicts previously known composite elements, but experiments to elucidate the nature of the predicted novel interaction between JUN and IKZF1 are **incomplete**. These experiments would benefit from more rigorous approaches using other databases to explore additional relevant data. The work will be of broad interest to those involved in dissecting the regulatory logic of mammalian enhancers and promoters.

**\*For correspondence:**
lip3@nhlbi.nih.gov (PL);
leonardw@nhlbi.nih.gov (WJL)

[†]These authors contributed equally to this work

## Abstract

Transcription factor partners can cooperatively bind to DNA composite elements to augment gene transcription. Here, we report a novel protein-DNA binding screening pipeline, termed Spacing Preference Identification of Composite Elements (SPICE), that can systematically predict protein binding partners and DNA motif spacing preferences. Using SPICE, we successfully identified known composite elements, such as AP1-IRF composite elements (AICEs) and STAT5 tetramers, and also uncovered several novel binding partners, including JUN-IKZF1 composite elements. One such novel interaction was identified at CNS9, an upstream conserved noncoding region in the human *IL10* gene, which harbors a non-canonical IKZF1 binding site. We confirmed the cooperative binding of JUN and IKZF1 and showed that the activity of an *IL10*-luciferase reporter construct in primary B and T cells depended on both this site and the AP1 binding site within this composite element. Overall, our findings reveal an unappreciated global association of IKZF1 and AP1 and establish SPICE as a valuable new pipeline for predicting novel transcription binding complexes.

## Introduction

Genome-wide studies exploring cooperative interactions of transcription factors (TFs), histone modifications, and higher-order chromatin conformation are critical for systematically evaluating gene expression mechanisms. Chromatin immunoprecipitation coupled with DNA sequencing (ChIP-seq) has been employed to map genome-wide DNA binding proteins of interest, advancing our understanding of genomics and epigenomics (*Barski et al., 2007*; *Roberson et al., 2007*), with important discoveries related to disease-associated transcriptional regulation (*Deardorff et al., 2012*; *Izumi et al., 2015*), tissue-specificity of epigenetic regulation (*Ernst et al., 2011*; *Mikkelsen et al., 2010*), and chromatin organization (*Sutani et al., 2015*). To extract biologically relevant information from ChIP-seq data, many computational tools, such as Model-based Analysis for ChIP-seq (MACS)(*Zhang et al., 2008*), MUltiScale enRichment Calling for ChIP-seq (MUSIC)(*Harmanci et al., 2014*), and Zero-Inflated Negative Binomial Algorithm (ZINBA)(*Rashid et al., 2011*), have been developed to understand transcription factor cooperation and interactions. Genomic regions that are significantly

enriched represent candidates for protein/DNA-binding sites that can be used to predict binding motifs (*Machanick and Bailey, 2011*), and ChIP-seq analysis can be integrated with other types of genomic assays, including RNA-seq gene expression profiling and long-distance chromatin interaction analyses to elucidate mechanisms of genomic function (*Li et al., 2012a*; *Li et al., 2017*) and gene ontology (*McLean et al., 2010*). Moreover, given a set of identified ChIP-seq peaks, motif enrichment analysis tools such as MEME, STREME, TF-COMB, MCOT, and TACO (*Machanick and Bailey, 2011*; *Bailey et al., 2009*; *Bailey, 2021*; *Levitsky et al., 2019*; *Jankowski et al., 2014*) can de novo discover the motif(s) corresponding to the TF's DNA binding domain. When binding to genomic DNA, some TFs may cooperatively bind and physically interact with specific partner TFs, and there often is preferred spacing between their binding sites (*Li et al., 2012a*; *Karczewski et al., 2011*; *Lin et al., 2012*; *Liu and Little, 1998*; *Jolma et al., 2015*). These cooperative interactions at composite elements are critical for the transcriptional regulation of genes and are used by cells to integrate diverse signals and potently drive transcription even at low concentrations of transcription factors. However, few computational tools to date can predict composite elements and their optimal spacing using transcription factor ChIP-seq data. SpaMo (space motif analysis) in MEME suite (*Bailey et al., 2009*) can infer interactions between a pair of specific TFs at near sites on the DNA sequence, however, it cannot systematically predict TF-TF composite elements genome-wide. We present a novel computational pipeline denoted 'SPICE,' which can predict genome-wide novel composite elements for pairs of transcription factors and their optimal motif spacing. SPICE is both versatile and efficient, capable of utilizing any ChIP-seq dataset to generate predictions that can be experimentally validated to confirm cooperative binding and its effects on gene expression. Our findings show that SPICE accurately predicts known binding partners and composite motifs, and in this study we demonstrate genome-wide JUN-IKZF1 interactions. SPICE is broadly applicable and serves as a powerful tool for uncovering a range of new transcription factor interactions and elucidating mechanisms underlying gene regulation.

## Results and discussion
### The SPICE pipeline predicts known and novel protein complexes

The SPICE schematic pipeline is summarized in *Figure 1A* and detailed in the Methods. First, we aligned the sequenced reads from ChIP-seq datasets to the reference genome and identified significant transcription factor binding sites (peaks) using MACS (*Zhang et al., 2008*). Second, we performed de novo motif analysis, designating the most enriched motif among the identified peaks as the primary motif. We retained only the peaks containing this primary motif and excluded sequences without matches. Third, we loaded secondary motifs using motif databases such as HOCOMOCO (*Kulakovskiy et al., 2018*), centered the primary motifs, and scanned for secondary motifs within 500 bp of the primary motif, focusing on the enrichment of known transcription factor DNA-binding motifs. Finally, we sorted the significant secondary motifs and their spacing, grouping redundant ones together. To visualize their interactions, we created heatmaps showing the interaction matrices of the primary and secondary motifs (see illustrative schematic in *Figure 1B*, where each red square indicates a predicted composite element containing the motifs at the corresponding primary and secondary motifs on the x- and y-axis, respectively). Based on the distance from the primary to the secondary motif, we displayed the spacing distributions as bar graphs highlighting the most significant spacing preferences (*Figure 1C*). To validate the predictive accuracy of SPICE, we used our previously published IRF4 ChIP-seq data in mouse pre-activated T cells to see if we could de novo re-discover AP-1/IRF4 composite elements (AICE, 5'-TGAnTCA/GAAA-3')(*Li et al., 2012b*). Indeed, SPICE successfully predicted these elements, including optimal spacing between AP1 and IRF4 as 0 or 4 bp (*Figure 1D*; *Figure 1—figure supplement 1A–B*), as previously reported (*Li et al., 2012b*), thus validating the pipeline. Furthermore, SPICE successfully predicted STAT5 tetramerization with an optimal spacing of 11–12 bps (*Figure 1—figure supplement 1C–D*), as previously reported (*Lin et al., 2012*). STAT1, STAT3, and STAT4 also have been reported to form tetramers (*Vinkemeier et al., 1996*; *Zhang and Darnell, 2001*; *Vinkemeier et al., 1998*), and indeed SPICE de novo predicted that these STATs also can form tetramers, whereas STAT2 was not predicted to form tetramers (*Supplementary file 1*), consistent with the lack of reports of such formation.

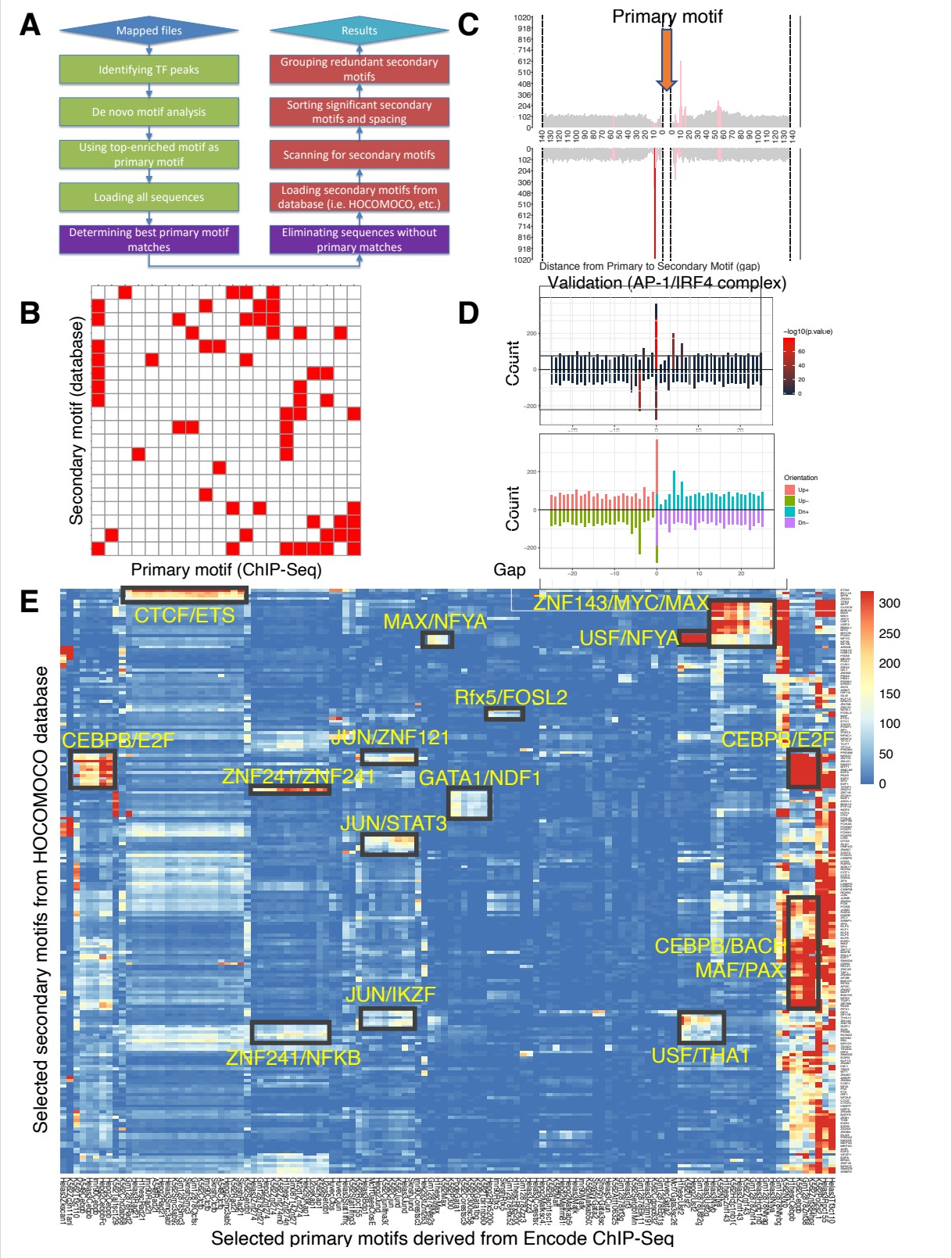

**Figure 1.** Spacing Preference Identification of Composite Elements (SPICE) predicts known and novel composite elements and spacing preferences. (**A**) Schematic step-by-step flowchart of the SPICE pipeline to predict transcription factor composite elements and their spacing preferences (see Methods for details). (**B**) Schematic of a heatmap depicting hypothetically enriched significant motif pairs (shown as solid red boxes). This is an interaction matrix of primary and secondary transcription factor motifs. It depicts the potential composite elements that can be predicted when

*Figure 1 continued on next page*

*Figure 1 continued*

a Chromatin immunoprecipitation coupled with DNA sequencing (ChIP-seq)-derived primary transcription factor motif region is compared with a database of secondary transcription factor motifs, based on the distribution of spacing between primary and secondary motifs. The red boxes indicate composite elements based on the primary and secondary motifs on the x- and y-axis, respectively. (C) Schematic bar graphs showing the spacing distribution of secondary motifs centered on a given primary motif, with bars highlighted in red, indicating the most preferred spacing. (D) SPICE successfully predicts and validates known composite elements. Shown is the validation of SPICE's ability to rediscover the known AP-1/IRF4 composite element (AICE). (E) Potential transcription factor composite elements identified from SPICE. Heatmap of the significant motif pairs, with at least one pair of motifs exhibiting an E-value $<1e^{-10}$. This 118×205 spatial interaction matrix using Transcription Factor Binding Sites (TFBS) from the ENCODE project represents a subset of the primary and secondary motifs shown in the interaction matrix shown in *Figure 1—figure supplement 2*. The x-axis represents the primary motifs derived from ChIP-seq libraries, and the y-axis indicates the transcription factor motifs from *Homo sapiens* Comprehensive Model Collection (HOCOMOCO) database. Potential novel combinatory complexes are highlighted in black boxes and yellow font (e.g. CTCF/ETS composite elements and JUN/IKZF composite elements).

The online version of this article includes the following figure supplement(s) for figure 1:

**Figure supplement 1.** Spacing Preference Identification of Composite Elements (SPICE) successfully predicts AICE, STAT5 tetramers, and CTCF/ETS complexes.

**Figure supplement 2.** Spacing Preference Identification of Composite Elements (SPICE) predicts transcription factor composite elements using the Encode Chromatin immunoprecipitation coupled with DNA sequencing (ChIP-seq) data.

**Figure supplement 3.** JUN-IKZF composite elements in GM12878 cells.

## SPICE can predict known and novel composite elements from the human Encode database

To evaluate SPICE's capability to predict novel transcription factor binding complexes, we used ChIP-seq data from the ENCODE project, which was standardized by the Analysis Working Group and utilized in a previous study (*Guo et al., 2012*), thus enabling us to conduct an integrated analysis of all ENCODE data types based on uniform processing. We evaluated the performance of SPICE using ENCODE ChIP-seq libraries from Transcription Factor Binding Sites (TFBS) generated by ENCODE/Stanford/Yale/USC/Harvard (SYUH), which comprise a total of 343 libraries in 20 different cell lines, with K562 human erythroleukemia, GM12878 EBV-transformed B cell, and HeLa S3 human cervical carcinoma cell lines being heavily used to generate these libraries. For each selected TF ChIP-seq library that contains a primary motif (*Figure 1—figure supplement 2*, x-axis), we extracted 500 bp of DNA sequence centered on the primary motif and searched for 401 known human transcription factor binding motifs using the *Homo sapiens* **Co**mprehensive **Mo**del **Co**llection database (HOCOMOCO, v11)(*Figure 1—figure supplement 2*, y-axis), and generated a 343×401 spatial interaction matrix, in which each cell was assigned an E-value indicating the significance of primary-secondary motif pairs (*Supplementary file 2*). The E-values were then log-transformed (-log10(E-value)) and shown as an interaction heatmap (*Figure 1—figure supplement 2*). As is evident, most of the motif pairs were not significant, suggesting that transcription factor composite elements are relatively rare events; in addition, some interactions that exist in some cell types may not be observed if they are not present in the cell lines that were used to generate the ENCODE datasets. After applying a filtering criterion that only included motif pairs with an E-value $<1e^{-10}$, the resulting interaction matrix had dimensions of 118x205 and exhibited reduced spatial complexity. Very interestingly, we could immediately identify multiple potential TF-TF composite elements (*Figure 1E*), which include putative associations of JUN with STAT3 and IKZF1, of CEBPβ with E2F, KLF, and BACH family members, and a range of others including ZNF241/NF-κB, GATA1/NDF1, and CTCF/ETS. Importantly, an association of JUN with STAT3 has been known (*Zhang et al., 1999*) and a functional CTCF-ETS1 interaction was recently reported (*Pham et al., 2019*), further validating the predictive power of SPICE. Regarding the CTCF-ETS association, SPICE successfully defined an optimal spacing of 8 bp for CTCF/ETS complexes (*Figure 1—figure supplement 1E–F*), something that was not defined in the earlier study (*Pham et al., 2019*) and thereby further underscoring the robustness of our pipeline. Interestingly, some TFs show significant interactions with a wide range of partner factors. These TFs are studied as 'Stripe' transcription factors, or 'universal stripe factors (USFs).' Examples include members of the KLF, EGR, and ZBTB families, which recognize overlapping GC-rich sequences across all analyzed tissues (*Zhao et al., 2022*).

## Most IKZF1 binding sites co-localize with JUN

We decided to focus on the putative JUN-IKZF1 given that both JUN and IKZF1 play critical roles in immune cells but have not been reported to physically associate or functionally cooperate. IKZF1, also known as IKAROS, plays a vital role in the hematopoietic system (*Mullighan et al., 2009*; *Ramírez et al., 2010*) and is a regulator of immune cell development in early B cells and T cells (*Nutt and Kee, 2007*; *Kleinmann et al., 2008*). JUN family factors are part of AP-1 complexes, forming homodimers or hetero-dimers with FOS family basic leucine zipper proteins and playing prominent roles in T cell biology (*Riera-Sans and Behrens, 2007*; *Carr et al., 2017*). By ChIP-seq, IKZF1, and JUN extensively co-localized in the human cell line GM12878 (*Figure 1—figure supplement 3A* left panel: binding intensity heatmap; right panel: co-localization Venn diagram), for example at multiple sites in the *TNFRSF8* and *IL10* loci. The *IL10* locus included a highly conserved CNS9 region (highlighted in a red box and labeled in *Figure 1—figure supplement 3B*), located approximately 9.1 kb upstream of the *IL10* promoter. Pairwise sequence alignment confirms the CNS9 region is highly conserved between humans and mouse (*Figure 1—figure supplement 3C*). BATF, a FOS family member, is also bound at these loci (*Figure 1—figure supplement 3B*), consistent with BATF's ability to partner with JUN in AICEs (*Li et al., 2012b*). To further characterize IKZF1-JUN complexes and validate their existence in vitro and in vivo, we performed ChIP-seq experiments with antibodies to IKZF1 and JUN in primary mouse B cells that were either untreated or treated with LPS or LPS + IL-21 for 3 hr. We used B cells given the importance of IKZF1 for B cell development (*Sellars et al., 2011*). When cells were stimulated with LPS + IL-21, we identified a total of 9197 and 14433 binding sites for IKZF1 and JUN, respectively (*Figure 2A*). The de novo motif analysis (HOMER) on the IKZF1 peaks (*Figure 2B*) indicated the most enriched motifs include ETS-like motif (ranks first) and EBF1-like motif (ranks second), which are comparable to the previously identified IKZF1 motifs (*Yoshida and Georgopoulos, 2014*), as well as an IRF-like motif (ranks third), etc., all of which ostensibly could be utilized by IKZF1 to bind to DNA. Strikingly, 8125 out of 9197 (88.3%) of IKZF1 peaks co-localized with JUN (*Figure 2C*), and the Reactome Pathway Analysis from the 8125 co-localized peaks suggested that the IKZF1-JUN complex might play broad biological roles as there were multiple enriched pathways, including 'Cytokine Signaling in Immune System,' 'Signaling by Interleukins,' TLR4-related cascades, and more. (*Figure 2D* and *Figure 2—figure supplement 1*). IKZF1 binding was induced when cells were treated with LPS or LPS + IL-21, while JUN binding was prominent only with LPS + IL-21 treatment. IKZF1 sites globally co-localized with JUN (*Figure 2E*), including at known regulatory elements, for example at the *Prdm1* and *Il10* genes (*Figure 2F*; *Li et al., 2012b*), which are critical for plasma cell differentiation in IL-10-producing B cells (*Wang et al., 2019*). Importantly, the CNS9 region was identified not only in the human *IL10* gene (*Figure 1—figure supplement 3B*) but also in the mouse *Il10* gene (*Figure 2F*), suggesting that this conserved region might be biologically important and dependent on the binding of both IKZF1 and JUN. Importantly, besides IKZF1 and JUN binding, other pioneer factors might also play vital roles and contribute to biological function by opening chromatin, thus facilitating IKZF1 and JUN binding to DNA, an interesting area for future investigation.

## Cooperative binding of IKZF1 and JUN at the *IL10* upstream region

To identify potential physical interactions between IKZF1 and JUN, we performed co-immunoprecipitation experiments in the MINO cell line, a human mantle cell lymphoma B cell line. We confirmed that both JUN and IKZF1 were expressed in these cells, as assessed by western blotting (*Figure 3A*). We have used antibodies to c-JUN (Cell Signaling Technologies, 9165 S) and JunD (Cell Signaling Technologies, D17G2). Although we could not co-immunoprecipiate IKZF1 with anti-JUN or anti-JunD, JUN was weakly co-precipitated by anti-IKZF1, suggesting these factors might form a complex (*Figure 3A*), consistent with their binding in proximity in a genome-wide fashion as shown in *Figure 2C and E*. The gene encoding IL-10 in both humans and mice is expressed in both T cells and B cells (*Saraiva and O'Garra, 2010*) and contains a canonical AP-1 motif and a nearby sequence (GTTGCAGTTTC) that is an 11 bp match for the variant third IKZF1 motif peak in *Figure 2B*. These motifs were located in the CNS9 region (~9.1 kb upstream of the *IL10* promoter) (*Figure 2F* and *Figure 1—figure supplement 3B*), which is a highly conserved non-coding region in the mouse and human genes (*Figure 1—figure supplement 3C*) and is critical for regulation of the *IL10* gene (*Li et al., 2012b*; *Lee et al., 2009*). We wished to assess whether this might be a variant IKZF1 motif and performed EMSAs using wild-type or mutant probes corresponding to CNS9 (*Figure 3B*; the GTTGCAGTTTC motif 3 region and AP1

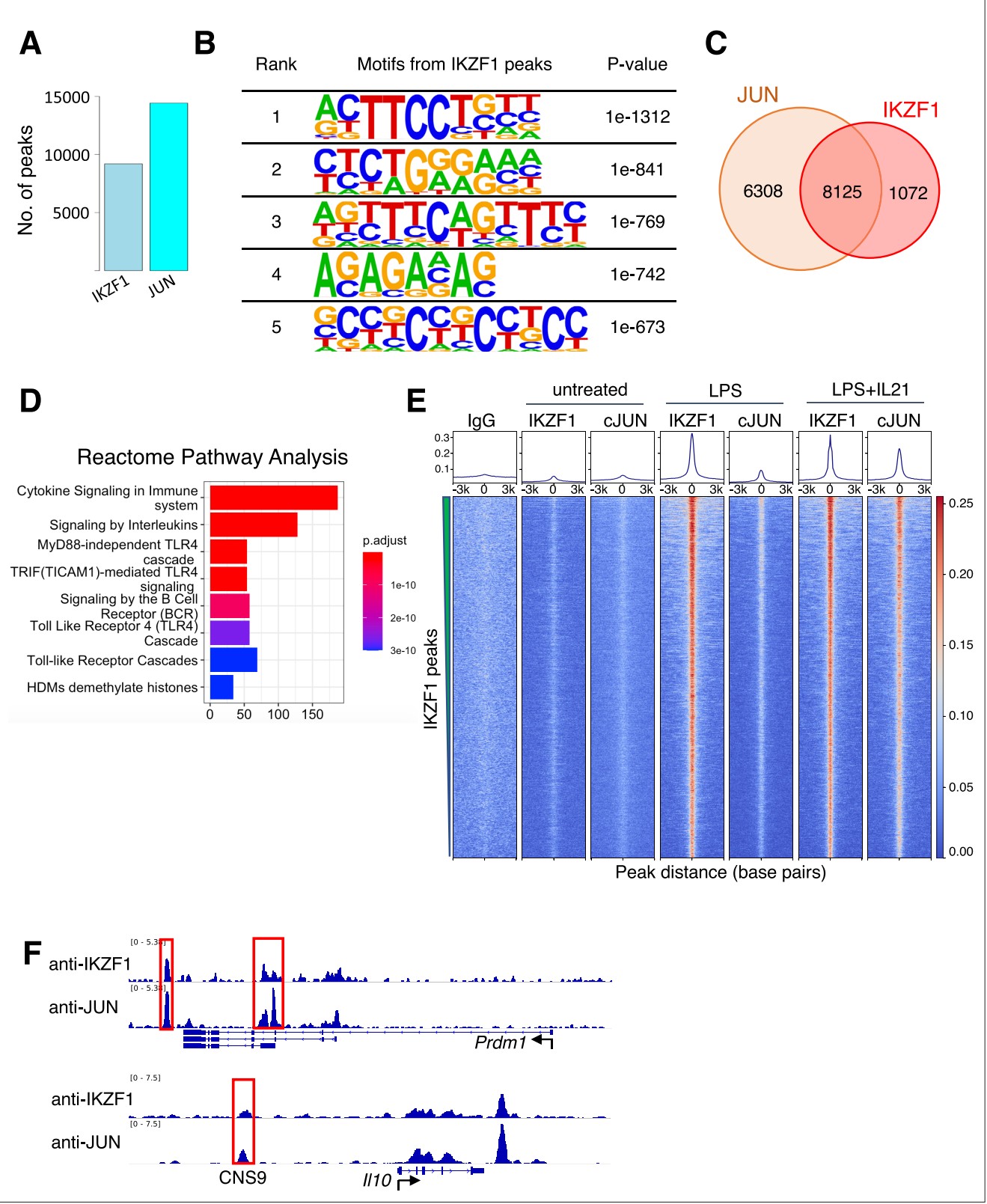

**Figure 2.** JUN co-localizes with IKZF1. (**A**) Number of Chromatin immunoprecipitation coupled with DNA sequencing (ChIP-seq) peaks identified for IKZF1 and JUN in experiments from primary mouse B cells that were treated with LPS + IL-21. (**B**) De novo motif discovery (HOMER) from IKZF1 peaks in primary mouse B cells treated with LPS + IL-21. The most significantly enriched motifs and their corresponding p-values are shown. (**C**) Venn diagram showing that most (88.3%) IKZF1 peaks co-localize with JUN. (**D**) Reactome pathways analysis from the IKZF1-JUN co-localized peaks reveal that

*Figure 2 continued on next page*

Figure 2 continued

cytokine-related pathways were most enriched, including 'Cytokine Signaling in Immune System,' 'Signaling by Interleukins,' and several TLR4-related pathways/cascades. (**D**) Heatmap showing that IKZF1 binding was potently activated by LPS or LPS + IL-21 treatment and that IKZF1 binds in proximity to JUN. Shown are normalized ChIP-seq signals ± 3 kb, centered on IKZF1 peak summits. (**E**) IGV browser file shows IKZF1 and JUN bound to the *Prdm1* and *Il10* genes. The red boxes indicate known regulatory elements. The position of a known upstream cis-regulatory element (CNS9) that controls *Il10* expression is indicated.

The online version of this article includes the following figure supplement(s) for figure 2:

**Figure supplement 1.** Gene-concept network depicts the linkages of genes and biological concepts of the top five Reactome Pathways that were enriched in IKZF1-JUN shared peaks.

---

motifs are underlined) and MINO nuclear extracts. A strong complex formed with the WT CNS9 probe (*Figure 3C*, lane 2), but the intensity was weaker when the variant IKZF1 sequence was mutated (lane 4) and abolished when the AP-1 motif (lane 6) or both motifs (lane 8) were mutated (*Figure 3C and D*). We next evaluated the presence of IKZF1 and JUN by super-shifting with antibodies. As expected, a strong complex was seen only in the presence of nuclear extract (*Figure 3E*, lane 2 vs 1). Neither mouse nor rabbit IgG affected the complex (lanes 3 and 4), but anti-IKZF1 consistently altered the

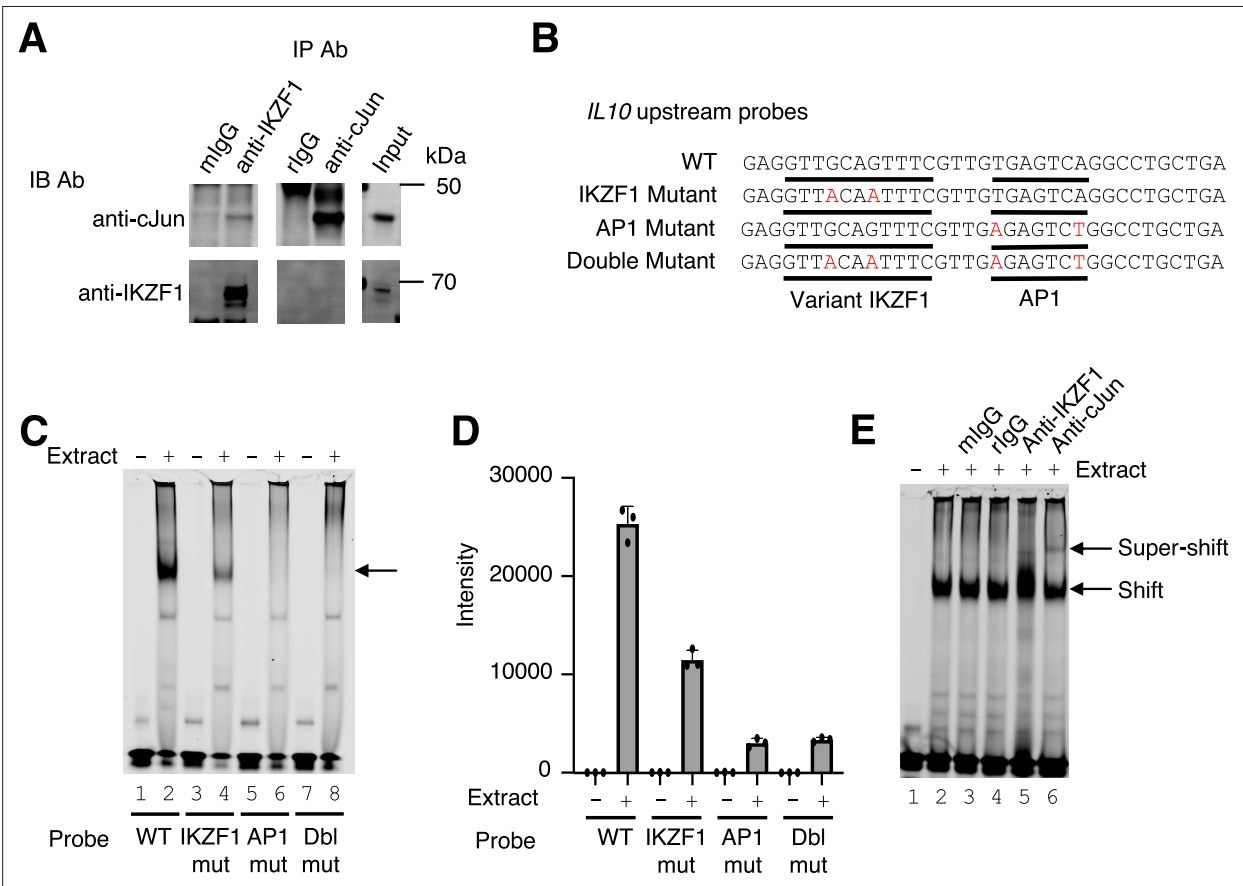

**Figure 3.** Cooperative binding of IKZF1 and JUN at an *IL10* upstream site, cis-regulatory element (CNS9). (**A**) Co-immunoprecipitation reveals IKZF1 physically associates with JUN. Nuclear protein lysates from $25 \times 10^6$ cells were used for each IP reaction. Protein extracts (input) were immunoprecipitated with antibodies to IKZF1, JUN, or normal IgG (mouse and rabbit) and resolved by SDS-PAGE. Western blotting was then performed with antibodies to IKZF1 or JUN. (**B**) Wild-type and mutant probes at the *IL10* CNS9 upstream site; variant IKZF1 and AP-1 motifs are underlined, with mutant nucleotides colored in red. (**C**) Representative EMSA with *IL10* upstream probes (wild-type, or IKZF1, AP-1, or IKZF1/AP-1 mutants; see panel B) and nuclear extracts from MINO cells. 100 nM of IR700-labeled probes were used per EMSA reaction. 5 µg of MINO nuclear extract was used in lanes 2, 4, 6, and 8; no nuclear extract was added in control lanes 1, 3, 5, and 7. (**D**) Average intensities from three independent EMSA experiments (one of which is shown in panel C). Band intensities were calculated using Image Studio software (LI-COR). (**E**) EMSA super-shifting was performed with 5 µg of nuclear extracts from MINO cells and mouse IgG, rabbit IgG, anti-IKZF1, and anti-JUN. Nuclear lysates were pre-incubated for 20 min on ice with 1 µg of the indicated antibodies prior to the addition of the probe. EMSAs were performed at least three times.

appearance of the shift and anti-JUN gave a discrete slower mobility super-shifted band, consistent with both IKZF1 and JUN being part of the complex (lanes 5 and 6).

## Both IKZF1 and JUN are critical for cooperative binding and transcriptional activation

We next performed EMSAs using the WT IR700 probe and 25, 50, 100, and 200 molar excess of cold competitor oligonucleotides corresponding to the WT or the variant IKZF1 mutant, AP-1 mutant, or double mutant CNS9 probes. The WT competitor dramatically reduced the binding activity and the variant IKZF1 mutant competitor moderately reduced the intensity of the shift, whereas AP-1 mutant or IKZF1/AP-1 double mutant oligonucleotides had little if any effect (*Figure 4A and B*). To determine whether IKZF1 and AP-1 proteins cooperatively bound to the CNS9 region, we used recombinant IKZF1 and cJUN/cFOS AP1 protein to perform EMSAs with wild-type and mutant IR700 probes corresponding to the *IL10* CNS9 site (data shown in *Figure 4C* and quantified in *Figure 4D*). There is a second band with faster mobility beneath the prominent band in the *Figure 4C* (EMSA experiment with recombinant IKZF1 and JUN), which perhaps was bound by IKZF1 since its intensity weakened when the IKZF1 site was mutated and was absent when only JUN was added. With the wild-type probe, no binding activity was observed with IKZF1 alone (*Figure 4C and D*, **lane 2**), but binding occurred with AP-1 (**lane 3**), and stronger binding occurred when both IKZF1 and AP1 proteins were added (**lane 4**), suggesting cooperative binding of these factors. Binding was not detected to the AP-1 mutant probe (**lane 8**) and was lower with the variant IKZF1 mutant probe (**lane 6**), suggesting that the AP-1 site is absolutely required for the cooperative binding and that the putative variant IKZF1 site had low affinity in its own right but cooperated with the AP1 site to allow cooperative binding of the factors. Correspondingly, the activity of the WT *IL10* luciferase reporter was potently induced by LPS in primary B cells compared to vector but not when IKZF1, AP-1, or both motifs were mutated (*Figure 4E*), suggesting both IKZF1 and AP-1 are important for transcriptional activation of this gene. However, since AP-1 comprises a group of homodimers or heterodimers of JUN/JUNB/BATF/FOS family members, the roles of the individual members remains uncertain, with the different members having potentially similar or even opposing actions. Besides B cells, IKZF1 has been shown to be a regulator of *Il10* gene expression in mouse T cells, in which *Il10* expression was significantly lower compared with wild-type T cells in response to TCR stimulation or during Th2 differentiation in *Ikzf1*[-/-] cells (*Umetsu and Winandy, 2009*). We, therefore, also analyzed activity in T cells. *IL10* reporter activity was potently induced, but the expression was diminished when the variant IKZF1 motif, or particularly the AP1 motif or both motifs were mutated (*Figure 4F*), demonstrating cooperative binding and transcriptional activation by IKZF1 and AP1 proteins in this setting as well. These results provide mechanistic insights into the actions of IKZF1 and underscore the power of the SPICE pipeline for identifying new transcription factor cooperative effects.

We have developed a novel computational pipeline called SPICE, which was designed to predict new transcription factor binding partners and identify optimal DNA spacing between motifs. Our pipeline is unique, and our findings are distinctive. SPICE systematically screens and predicts novel protein-protein binding complexes, including the newly discovered IKZF1-JUN composite element, yielding distinctive and valuable biological insights and validations. In this report, we utilized the SPICE pipeline to analyze the ENCODE database, successfully predicting both a range of known interactions and numerous novel interactions. Although newer and updated databases like ReMap (*Hammal et al., 2022*), which includes over 8000 ChIP-seq datasets, are available, our fundamental conclusion regarding the IKZF1-JUN composite element would not be altered even if a larger database were used. Here, we focused on JUN-IKZF composite elements, an association not previously demonstrated. ChIP-seq analysis demonstrated that IKZF1 (Ikaros) co-localizes with JUN in a genome-wide manner, and that the major two motifs identified were canonical IKZF1 motifs, but that non-canonical variant motifs were also identified, including at the *IL10* gene locus. It is worth noting that because IKZF1 is closely related to IKZF2 (Helios) and IKZF3 (Aiolos), it is conceivable that one or both of these factors might also co-localize with JUN (*Georgopoulos, 2017*). Notably, an independent study that cited our ChIP-seq data reported in this study confirmed that IKZF1 and IKZF3 can directly bind to AP-1 family members (*Goh et al., 2024*). In that study, the deletion of both *Ikzf1* and *Ikzf3* in NK cells led to a significant reduction in Jun/Fos expression and a complete loss of peripheral NK cells. This independent validation supports our findings and further suggests that IKZF1 and JUN

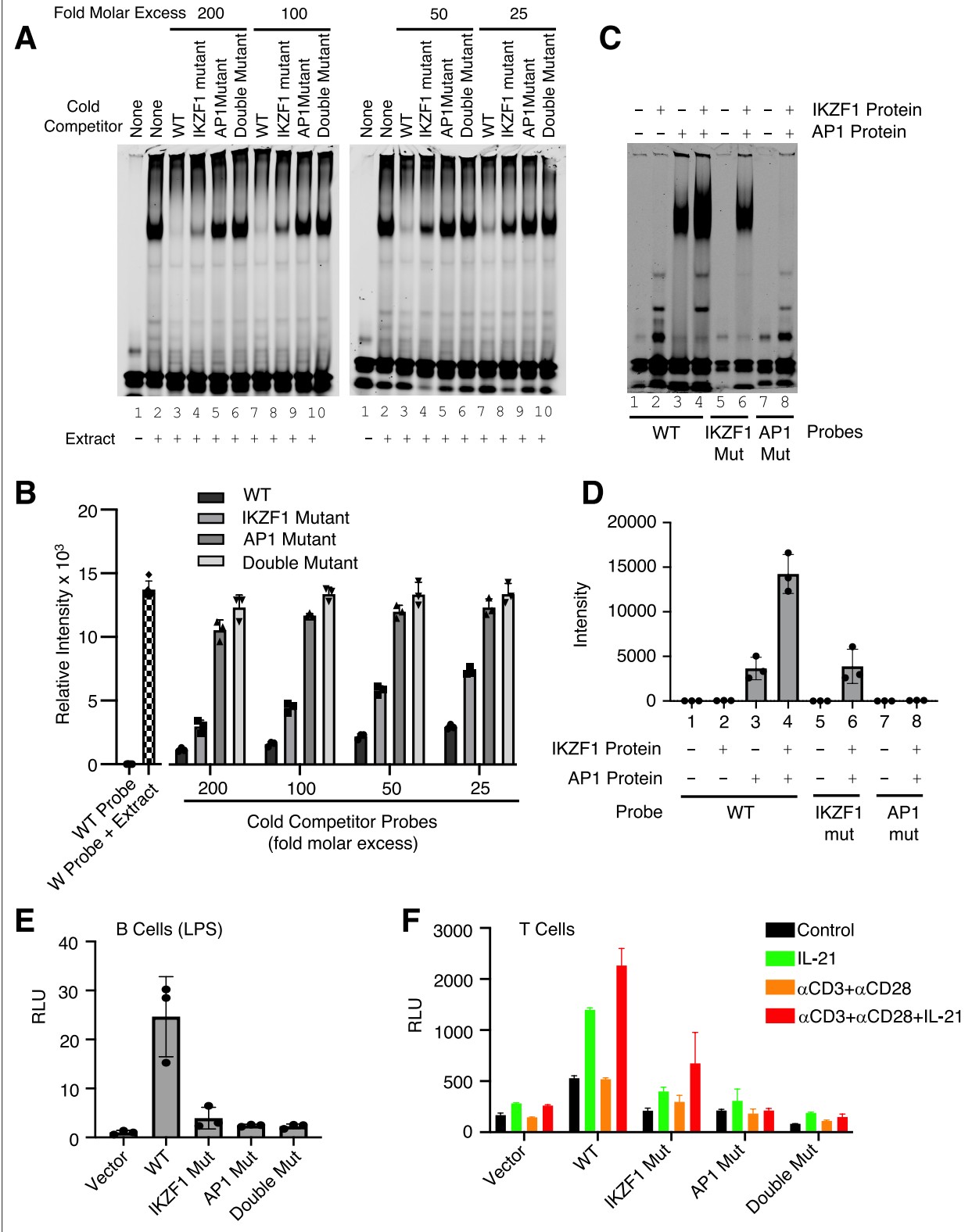

**Figure 4.** Both IKZF1 and JUN are critical for cooperative binding and transcriptional activation. (**A, B**) Cooperative binding of IKZF1 and AP1. EMSA was performed with a wild-type (WT) *IL10* probe corresponding to cis-regulatory element (CNS9) and nuclear extracts from MINO cells stimulated with LPS + IL-21. Un-labeled 'cold' wild-type or mutant double-stranded oligonucleotides (25, 50, 100, or 200-fold molar excess relative to the IR700-labeled probes) were added to 5 µg nuclear extracts (prepared from MINO cells treated with LPS + IL-21) prior to the addition of 100 nM of WT IR700-labeled probe. Shown are a representative EMSA (**A**) and the summary of relative band intensities from three independent experiments (**B**). Relative

*Figure 4 continued on next page*

*Figure 4 continued*

intensities were calculated using Image Studio software (LI-COR). (**C, D**) EMSA using human recombinant AP-1 or IKZF1 with *IL10* CNS9 upstream probes (wild-type, or IKZF1/AP-1 mutants). 300 ng of recombinant AP1 protein (150 ng of each cJUN and cFOS) and 1 μg of recombinant IKZF1 were used as indicated. Band intensities from three independent experiments are shown in panel D. (**E**) WT or mutant reporter constructs were transfected via electroporation into mouse primary B cells pre-stimulated with LPS for 24 hr. After electroporation, cells were treated with LPS for 24 hr before dual luciferase activity was measured (n=3; mean ± S.D). (**F**) WT or mutant reporter constructs were transfected into mouse primary pre-activated CD8+ T cells and the cells were then treated as indicated for 24 hr (n=3; mean ± S.D). RLU, relative light units.

can functionally cooperate. In our study, we could coprecipitate JUN with anti-IKZF1, and moreover, we found functional cooperation between IKZF1 and AP1 motifs. Specifically, we found that IKZF1 and JUN not only cooperatively bound to the CNS9 region in the *IL10* locus but that both IKZF1 and AP1 motifs were critical for transcriptional activity of a reporter construct in both B and T cells. While our primary focus was on the *IL10* locus, we also demonstrated genome-wide co-localization of these factors. Reactome pathway analysis associates the genes they co-localize with across various pathways, including, but not limited to, cytokine signaling pathways. (***Figure 2***); Therefore, the physiological significance of transcription factor cooperation may be broad, potentially influencing various cellular functions and disease contexts. The discovery of JUN-IKZF1 complexes highlights the effectiveness of our computational approach and specifically of the SPICE pipeline in predicting novel transcription factors binding complexes and their optimal spacing. This provides a valuable new way to discover composite elements that contribute to a highly specific pattern of gene transcriptional regulation and to better understand gene regulatory networks involved in the immune responses.

## Materials and methods
### SPICE pipeline
All mapped files of TFBS were downloaded from ENCODE project (http://hgdownload.soe.ucsc.edu/goldenPath/hg19/encodeDCC/wgEncodeSydhTfbs/), from a collection of libraries generated by Stanford/Yale/USC/Harvard (SYUH). The following are the step-by-step analysis corresponding to the flowchart in ***Figure 1A***.

 '*Identity TF peaks:*' MACS 1.4.2 (8) was used to identify significant peaks for each transcription factor using input IgG as control, *p-value* threshold was set as $1e^{-10}$. Alternatively, one can directly download SYUH pre-defined peaks-- i.e., SydhTfbsGm12878JundIggrabPk.narrowPeak. '*De novo motif analysis:*' MEME (multiple EM for motif elicitation) was used to 'de novo' discover consensus primary motifs from the 1000 most significant TF binding sites (sorted by *-log10(p-value)* in peak file). '*Using top-enriched motif as primary motif:*' The top-enriched motif (sorted by *e-value*, which is based on its log-likelihood ratio, width, sites, the background letter frequencies) was selected as the primary motif. '*Loading all sequences*,' '*Determining best primary motif matches*,' and '*Eliminating sequences without primary matches:*' If there was indeed a primary motif from a particular TF ChIP-seq, we extracted 500 bp DNA sequences centered on TF peak summits and only kept sequences containing primary motif matches and eliminated sequences without primary matches. '*Loading secondary motifs from database*' and '*Scanning for secondary motifs:*' We performed a spacing distribution analysis using SpaMo from MEME suite (***Bailey et al., 2009***) from DNA sequences containing primary motif to search for the strongest secondary motif from a motif database (HOCOMOCO, v11) to identify significantly enriched spacings. The relative spacings of the primary and secondary motifs in all identified ChIP-seq peaks were counted and the probability of each spacing was calculated. A *p-value* was assigned based on the Binomial Test, using the number of observed secondary spacings falling into the given 1 bp bin, adjusted for the number of bins tested. '*Sorting significant secondary motifs and spacing*' and '*Grouping redundant secondary motifs:*' After all the calculations were done, SPICE ranked the non-redundant secondary motifs with the best spacing, or gap (e.g. 4 bp) that passed the significance threshold (p-value <0.01) in order of significance. Similar secondary motifs belonging to the same cluster (e.g. ETS1 and ELF1 belong to the same ETS cluster) were grouped together and ranked in order of *p-value*. We performed the above step-by-step analysis iteratively for each individual TF ChIP-seq and generated the final result of interaction sparse matrix to plot in heatmap. The batch and R scripts associated with SPICE analysis, including de novo motif analysis, SpaMo analysis, and visualization of transcription factor composite elements are deposited into GitHub(copy archived

at *Li et al., 2024*) which is a developer platform that allows researchers, reviewers, and developers to create, store, manage and share their code.

## Cell culture and reagents

MINO cells were cultured in RPMI-1640 medium (Gibco, 11875–003) containing 10% fetal bovine serum (FBS, Gemini Bio-Products, 100–106) and 1% penicillin-streptomycin (Gemini Bio-Products, 400–109). Mouse splenic B cells were isolated using the EasySep Mouse B cell isolation Kit (STEMCELL technologies, 19854) according to the manufacturer's protocol. Primary mouse B cells were cultured in RPMI-1640 medium containing 10% FBS, 1% penicillin-streptomycin, and 50 µM 2-mercaptoethanol (Gibco, 21985023). LPS (*Escherichia coli* O111:B4) was purchased from Sigma-Aldrich (L3012). Recombinant human and mouse IL-21 protein was purchased from R & D systems.

## Mice

Wild-type 6–10 wk old female C57BL/6 background mice were purchased from Charles River Laboratories (strain #556). All experiments with mice were conducted under protocols approved by the NHLBI Animal Care and Use Committee. NIH guidelines for the use of animals in intramural research were followed.

## Electrophoretic mobility shift assays

MINO cells were treated with LPS (1 µg ml$^{-1}$) and IL-21 (100 ng ml$^{-1}$) for 3 hr. 10$^7$ cells were washed with 1 ml of ice-cold PBS and resuspended in 1 ml of ice-cold lysis buffer (10 mM HEPES pH 7.9, 10 mM KCL, 0.1 mM EDTA, 0.1 mM EGTA, 1 mM DTT, and protease inhibitors). After 10 min of incubation on ice, 32 µl of 10% NP-40 was added and vortexed for 10 s. The samples were centrifuged at 13,000 rpm for 30 s, and the supernatants were discarded. Pellets were washed with 1 ml of ice-cold lysis buffer and centrifuged at 13,000 rpm for 30 s, supernatants were discarded, and 100 µl of nuclear extraction buffer (20 mM HEPES pH 7.9, 0.4 M NaCl, 1 mM EDTA, 1 mM EGTA, 1 mM DTT and protease inhibitors) was added to each tube and vortexed. The samples were incubated on ice for 30 min and spun out at 13,000 rpm for 10 min. The supernatant (nuclear extract) was transferred to a fresh tube and the pellet was discarded. The protein concentration was measured using the Pierce BCA protein assay kit – reducing agent compatible (Cat. No. 23250). IRDye 700-labeled and non-labeled probes were purchased from IDT DNA technologies. The EMSA reactions were set up according to the manufacturer's protocol (LI-COR, P/N: 829–07910). 5 µg of MINO nuclear extract was used per reaction. For super-shift assays, the nuclear extract was pre-incubated for 20 min on ice with anti-cJUN rabbit mAb (clone 60A8) from Cell Signaling technologies (#9165 S) or with anti-IKZF1 (clone IK14), provided by Dr. Katia Georgopoulos, Massachusetts General Hospital, Boston, MA. Recombinant human cJUN (ab84134), cFOS (ab56280), and IKZF1 (ab169877) proteins were purchased from Abcam. Equimolar concentrations of recombinant cJUN and cFOS proteins were mixed together and incubated on ice for 20 min to generate recombinant AP1 protein for EMSA. 300 ng of recombinant AP1 and 1 µg of recombinant IKZF1 were used per EMSA reaction. 10 X orange loading dye was purchased from LI-COR (P/N 927–10100). The samples were run on a 4–12% TBE gel (Invitrogen, EC62352BOX) in 0.5 X TBE buffer (KD medical, RGE-3330).

## Co-immunoprecipitation assays

MINO cells were stimulated for 3 hr with LPS (1 µg ml$^{-1}$) and IL-21 (100 ng ml$^{-1}$), and nuclear co-IP was performed according to the manufacturer's protocol (Active Motif, 54001). 25×10$^6$ cells were used per IP reaction. Protein lysates were immunoprecipitated with antibodies to cJUN (Cell Signaling Technologies, 9165 S), IKZF1 (Sigma-Aldrich, MABE913), normal mouse IgG (Santa Cruz, sc-2025) and normal rabbit IgG (Cell Signaling Technologies, 2729 S). 5 µg of antibody was used per reaction. The protein-antibody complex was captured using Pierce protein A/G magnetic beads (88802). NuPage LDS sample buffer (4 X) (Invitrogen, NP0007) and NuPage sample reducing agent (10 X) (Invitrogen, NP0004) were used to prepare the IP samples for denaturing western blot. The samples were loaded onto NuPage 4–12% bis-tris gels (Invitrogen, NP0335BOX). After gel electrophoresis, the proteins were transferred onto a PVDF Low fluorescence membrane (Bio-Rad, 1620261). The membranes were blocked for 1 hr at room temperature with 5% skim milk in Tris-buffered Saline (TBS). After washing three times with TBS, the membrane was incubated overnight at 4 °C with primary antibodies to cJUN

(Cell Signaling Technologies, 9165 S), or IKZF1 (Sigma-Aldrich, MABE913). Blots were then washed and probed with secondary antibodies: IRDye 680RD Goat anti-rabbit IgG (LI-COR, P/N 926–68071) or IRDye 800CW Goat anti-mouse IgG (LI-COR, P/N 926–32210) at 1:10000 dilution. The membranes were imaged using ODYSSEY CLx (LI-COR).

## Luciferase reporter assays

Mouse splenic B cells were pre-stimulated with LPS (1 µg ml$^{-1}$) for 24 hr. $5 \times 10^6$ cells were electroporated with 1 µg of NanoLuc pNL3.1 (Promega, N1031) vector and 0.2 µg of control vector, pGL4.S4 luc TK (Promega) using the P3 Primary cell 4D-nucleofector X kit S from Lonza (V4XP-3032) according to the manufacturer's protocol. The wild-type and mutant oligos were cloned into the pNL3.1 vector using XhoI and HindIII restriction sites. After electroporation, the cells were stimulated with 1 µg ml$^{-1}$ LPS for 24 hr. Dual luciferase reporter activity was measured using the Nano-Glo Dual-Luciferase Reporter assay (Promega, N1630). The luciferase activity data shown is relative to the control pGL4.S4 luc TK activity. Primary mouse CD8$^+$ T cells were purified from spleens from WT C57BL/6 mice. Cells were activated for 24 hr with anti-CD3 (2 µg ml$^{-1}$) and anti-CD28 (1 µg ml$^{-1}$). Cells ($0.5 \times 10^6$) were then transfected with 1 µg nanoluciferase reporter constructs by nucleofection and stimulated immediately for 24 hr with IL-21 or anti-CD3 + anti-CD28 or the combination of IL-21 with anti-CD3 + anti-CD28. Luciferase activity was measured 24 hr later and expressed as RLU relative to the control plasmid.

## ChIP-seq library preparation

For ChIP-seq experiments, 8–10 million B cells were treated with either LPS (1 µg ml$^{-1}$) or LPS +mouse IL-21 (100 ng ml$^{-1}$) for 3 hr. Immunoprecipitation was performed overnight at 4 °C using antibodies to cJUN (Cell Signaling Technologies, 9165 S), IKZF1 (Active Motif, 39291), normal mouse IgG (Santa Cruz, sc-2025), or normal rabbit IgG (Cell Signaling Technologies, 2729 S). ChIP-seq DNA libraries were prepared with the KAPA LTP Library Preparation Kit and barcoded with NEXTflex DNA barcodes, quantified, and sequenced on an Illumina HiSeq 3000 system.

## Bioinformatics analyses

Sequenced reads (50 bp, single end) were obtained with the Illumina CASAVA pipeline and mapped to the mouse genome mm10 (GRCm38, December 2011) using Bowtie 2.2.6. Only uniquely mapped reads were retained. The mapped outputs were converted to browser-extensible data files, which were then converted to binary tiled data files (TDFs) using IGVTools 2.4.13 for viewing on the IGV browser (https://software.broadinstitute.org/software/igv/). TDFs represent the average alignment or feature density for a specified 20 bps window size across the genome. Peak calling was performed by MACS 1.4.2 (8) relative to an IgG library as input control and the p-value threshold was set as 1e$^{-10}$. Only non-redundant reads were analyzed for peak calling. ChIP-seq binding intensity heatmaps were analyzed and visualized using deepTools. ChIP-seq co-localization analysis was performed using the R package 'ChIPseeker,' and Reactome pathway analysis was performed using the R package 'ClusterProfiler'.

## Acknowledgements

This work was supported by the Division of Intramural Research, National Heart, Lung, and Blood Institute (NHLBI). Next generation sequencing was performed at the NHLBI Sequencing Core.

# Additional information

### Competing interests

Peng Li: is an employee of Amgen Inc. The other authors declare that no competing interests exist.

## Funding

| Funder | Grant reference number | Author |
|---|---|---|
| National Heart, Lung, and Blood Institute | Z99 HL999999 | |

The funders had no role in study design, data collection and interpretation, or the decision to submit the work for publication.

## Author contributions

Peng Li, Conceptualization, Data curation, Software, Formal analysis, Supervision, Validation, Investigation, Visualization, Methodology, Writing – original draft, Project administration, Writing – review and editing; Sree Pulugulla, Formal analysis, Validation, Visualization, Methodology, Writing – original draft, Writing – review and editing; Sonali Das, Formal analysis, Validation, Visualization; Jangsuk Oh, Rosanne Spolski, Validation, Investigation; Jian-Xin Lin, Investigation; Warren J Leonard, Resources, Supervision, Funding acquisition, Investigation, Writing – original draft, Project administration, Writing – review and editing

## Author ORCIDs

Peng Li ![ORCID] https://orcid.org/0000-0002-4721-5936
Warren J Leonard ![ORCID] https://orcid.org/0000-0002-5740-7448

Reviewer #1 (Public Review): https://doi.org/10.7554/eLife.88833.2.sa1
Reviewer #2 (Public Review): https://doi.org/10.7554/eLife.88833.2.sa2
Reviewer #3 (Public Review): https://doi.org/10.7554/eLife.88833.2.sa3
Author response https://doi.org/10.7554/eLife.88833.2.sa4

# Additional files

## Supplementary files

Supplementary file 1. Spacing Preference Identification of Composite Elements (SPICE) can predict possible tetramers for STAT family member proteins. SPICE is capable of identifying canonical STAT binding motifs, also known as interferon gamma-activated sequence (GAS) motifs, and then predicts STAT tetramer formation and the optimal spacing for tetramer formation for STAT1, STAT3, STAT4, STAT5A, and STAT5B, whereas STAT2 was predicted to not form tetramers. The various STAT family members were activated in the indicated cell types by the indicated cytokine, Chromatin immunoprecipitation coupled with DNA sequencing (ChIP-seq)-derived canonical GAS motifs, percentage of GAS motifs, likelihood of tetramer formation, and predicted optimal spacing were determined.

Supplementary file 2. The spatial interaction matrix of primary motifs from Chromatin immunoprecipitation coupled with DNA sequencing (ChIP-seq) libraries and secondary motifs from the *Homo sapiens* Comprehensive Model Collection database (HOCOMOCO) database. Each row represents one of the 401 motifs from the HOCOMOCO database, and each column is derived motifs from 343 TFBS ChIP-seq libraries. Each cell has an E-value indicating the significance of motif pairs, which is the lowest p-value of any spacing of the secondary motif times the number of secondary motifs. The value of 1 indicates there is no interaction between motif pairs.

MDAR checklist

## Data availability

Sequencing data have been deposited in GEO under accession code GSE230035.

The following dataset was generated:

| Author(s) | Year | Dataset title | Dataset URL | Database and Identifier |
|---|---|---|---|---|
| Li P, Pulugulla SH, Das S, Oh J, Spolski R, Lin J, Leonard WJ | 2023 | A new pipeline SPICE identifies novel JUN-IKZF1 composite elements | https://www.ncbi.nlm.nih.gov/geo/query/acc.cgi?acc=GSE230035 | NCBI Gene Expression Omnibus, GSE230035 |

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
