## [Editor Report · eLife assessment]

This **valuable** study presents a screening pipeline (SPICE) for detecting DNA motif spacing preferences between TF partners. SPICE predicts previously known composite elements, but experiments to elucidate the nature of the predicted novel interaction between JUN and IKZF1 are **incomplete**. These experiments would benefit from more rigorous approaches using other databases to explore additional relevant data. The work will be of broad interest to those involved in dissecting the regulatory logic of mammalian enhancers and promoters.

---

## [Referee Report · Reviewer #1 (Public Review)]

The authors report a new bioinformatics pipeline ("SPICE") to predict pairwise cooperative binding-sites based on input ChIP-seq data for transcription factor (TF)-of-interest, analyzed against DNA-binding sites (DNA motifs) in a database (HOCOMOCO). The pipeline also predicts the optimal distance between the paired binding sites. The pipeline correctly predicted known/reported transcription factor cooperations, and also predicted new cooperations, not yet reported in literature. The authors choose to follow up on the predicted interaction between Ikaros and Jun. Using ChIP-seq in mouse B cells, they show extensive overlap in binding regions between Ikaros and Jun in LPS+IL21 stimulated cells. In a human B-lineage cell line (MINO) they show that anti-Ikaros Ab can co-immunoprecipitate Jun protein, and that the MINO cell extracts contain protein(s) that can bind to the CNS9 probe (conserved region upstream of IL10 gene), and that binding is lost upon mutation of two basepairs in the AP1 binding motif, and reduced upon mutation of two basepairs in the non-canonical Ikaros binding motif. Part of this protein complex is super-shifted with an anti-Jun antibody, and more DNA is shifted with addition of an anti-Ikaros antibody.

The authors perform EMSA showing that recombinant Jun can bind to the tested DNA-region (IL10 CNS9) and that addition of recombinant Ikaros (or anti-Ikaros antibody in Fig 3E) can enhance binding (increase amount of DNA shifted). The authors lastly show that the IL10 CNS9 DNA region can enhance transcription in B- and T-cells with a luciferase reporter assay, and that 2 bp mutation of the Ikaros or Jun DNA motifs greatly reduce or abolish this activity.

This is interesting work, with two main contributions: The SPICE pipeline (if made available to the scientific community), and the report of interaction between Ikaros and Jun. However, the distinction between DNA motifs, and the proteins actually binding and having a biological function, should be made clear consistently throughout the manuscript. The same DNA motifs can be bound by multiple factors, for instance within transcription factor families with highly homology in the DNA-binding regions of the proteins.

Some specific points:

SPICE: It is unclear if this is uploaded somewhere to be available to the scientific community.

It was unclear if Ikaros-Jun interaction was initially found from primary Jun ChIP-seq (and secondary Ikaros motif from HOCOMOCO) or from primary Ikaros CHIP-seq (and secondary Jun motif from HOCOMOCO). And - what were the two DNA motifs (primary and secondary, and their distance) from the SPICE analysis?

Authors have mostly careful considerations and statements. One additional comment is that binding does not equal function (Fig 2D), and that opening of chromatin (by any other factor(s)) can give DNA-binding factors (like Ikaros and Jun) the opportunity to bind, without functional consequence for the biological process studied.

Figure 2E: Ikaros is reported to be expressed at baseline in murine B cells, yet the Ikaros ChIP-seq in unstimulated cells had what looks to be no significant or low peaks. LPS stimulation induced strong Ikaros ChIP-seq signal. A western blot showing the Ikaros protein levels in the 3 conditions could help understand if the binding pattern is due to protein expression level induction. Similar for Jun (western in the 3 conditions), which seemed to mainly bind in the LPS+IL21 condition. Furthermore, as also suggested below, tracks showing Ikaros and Jun binding from all conditions (unstimulated, LPS only and LPS+IL21 stimulated cells), at select genomic loci, would be helpful in illustrating this difference in signal between the different cell conditions. This is relevant in regards to the point of cooperativity of binding.

ChIP-seq in mouse B cells showed that Ikaros bound strongly in LPS stimulated cells, in the (relative) absence of Jun binding (Fig. 2C). However, in EMSA (Fig 3C), there is no binding when the AP1 site is mutated, and the authors describe this as Ikaros binding site. What does the Ikaros binding look like at this genomic location in LPS (only) stimulated cells? The authors could show the same figure as in Fig 2F but show Ikaros and Jun ChIP-seq tracks at IL10 CNS9 locus from all conditions to compare binding in unstimulated, LPS and LPS+IL21 cells.

Also: How does this reconcile with the luciferase assay in Fig 4E, where LPS (only) stimulation is used, which in Fig 2E only/mainly induced Ikaros, and not Jun ChIP-seq signal (while EMSA indicate Ikaros cannot bind the site alone, but can enhance Jun-dependent binding).

Comment on statements in results section: The luciferase assays in B and T cells do not demonstrate the role of the proteins Ikaros or Jun directly (page 10, lines 208 and surrounding text). The assay measures an effect of the DNA sequences (implying binding of some transcription factor(s)), but does not identify which protein factors bind there.

Lastly, the authors only discuss Ikaros (using the term IKZF1 which is the gene symbol for the Ikaros protein). There are other Ikaros family members that have high homology and that are reported to bind similar DNA sequences (for instance Aiolos and Helios), which are expressed in B-cells and T-cells. A discussion of this is of relevance, as these are different proteins, although belonging to the same family (the Ikaros-family) of transcription factors. For instance, western for Aiolos and Helios will likely detect Aiolos in the B cells used, and Helios in the T cells used.

---

## [Referee Report · Reviewer #2 (Public Review)]

The study is performed with old tool Spamo (12 year ago), source data from Encode (2010-2012), even peak caller tool version MACS is old ~ 2013. De novo motif search tool is old too (new one STREME is not mentioned). Any composite element search tool published for the recent 12 years are not cited, there are some issues in data analysis in presentation. Almost all references are from about 8-10 year ago (the most recent date is 2019)

The title is misleading

Instead of

A new pipeline SPICE identifies novel JUN-IKZF1 composite elements

It should be written as

Application of SpaMo tool identifies novel JUN-IKZF1 composite elements

It reflects the pipeline better but honestly shows that the novelty is missed.

The study was performed on too old data from ENCODE, authors mentioned 343 Encode ChIP-Seq libraries, but authors even did not care even about to set for each library the name of target TF (Figure 1E, Figure S2, Table 2).

---

## [Referee Report · Reviewer #3 (Public Review)]

The authors of this study have designed a novel screening pipeline to detect DNA motif spacing preferences between TF partners using publicly available data. They were able to recapitulate previously known composite elements, such as the AP-1/IRF4 composite elements (AICE) and predict many composite elements that are expected to be very useful to the community of researchers interested in dissecting the regulatory logic of mammalian enhancers and promoters. The authors then focus on a novel, SPICE predicted interaction between JUN and IKZF1, and show that under LPS and IL-21 treatment, JUN and IKZF1 in B cells have significant overlap in their genomic localization. Next, to know whether the two TFs physically interact, a co-immunoprecipitation experiment was performed. While JUN immunoprecipitated with an anti-IKZF1 antibody, curiously IKZF1 did not immunoprecipitate with an anti-JUN antibody. Finally, EMSA and luciferase experiments were performed to show that the two TFs bind cooperatively at an IL20 upstream probe.

Major strengths:

1. SPICE was able to recapitulate previously known composite elements, such as the AP-1/IRF4 composite elements (AICE).

2. Under LPS and IL-21 treatment, JUN and IKZF1 in B cells have significant overlap in their genomic localization. This is very good supporting evidence for the efficacy of SPICE in detecting TF partners.

Major weaknesses:

1. The authors fail to convincingly show that IKZF1 and Jun physically interact. A quantitative measurement of their interaction strength would have been ideal.

2. The super-shift experiment to show that the proteins bound to their EMSA probe were indeed IKZF1 and JUN are not very convincing and would benefit from efforts to quantify the shift (Figure 3E). Nuclear extracts from cells with single or double CRISPR knock outs of the two TFs would have been ideal.

3. There is a second band beneath the more prominent band in the EMSA experiment with recombinant IKZF1 and JUN (Figure 4C). This second band is most probably bound by IKZF1 because it becomes weaker when the IKZF1 site is mutated and is completely absent when only JUN is added. This is completely ignored by the authors. Therefore, experiments with EMSA fail to convincingly show that IKZF1 and Jun bind cooperatively. They could just as well bind independently to the two sites.

---

## [Author Response]

**Reviewer #1 (Public Review):**
The authors report a new bioinformatics pipeline ("SPICE") to predict pairwise cooperative binding-sites based on input ChIP-seq data for transcription factor (TF)-of-interest, analyzed against DNA-binding sites (DNA motifs) in a database (HOCOMOCO). The pipeline also predicts the optimal distance between the paired binding sites. The pipeline correctly predicted known/reported transcription factor cooperations, and also predicted new cooperations, not yet reported in literature. The authors choose to follow up on the predicted interaction between Ikaros and Jun. Using ChIP-seq in mouse B cells, they show extensive overlap in binding regions between Ikaros and Jun in LPS+IL21 stimulated cells. In a human B-lineage cell line (MINO) they show that anti-Ikaros Ab can co-immunoprecipitate Jun protein, and that the MINO cell extracts contain protein(s) that can bind to the CNS9 probe (conserved region upstream of IL10 gene), and that binding is lost upon mutation of two basepairs in the AP1 binding motif, and reduced upon mutation of two basepairs in the non-canonical Ikaros binding motif. Part of this protein complex is super-shifted with an anti-Jun antibody, and more DNA is shifted with addition of an anti-Ikaros antibody.The authors perform EMSA showing that recombinant Jun can bind to the tested DNA-region (IL10 CNS9) and that addition of recombinant Ikaros (or anti-Ikaros antibody in Fig 3E) can enhance binding (increase amount of DNA shifted). The authors lastly show that the IL10 CNS9 DNA region can enhance transcription in B- and T-cells with a luciferase reporter assay, and that 2 bp mutation of the Ikaros or Jun DNA motifs greatly reduce or abolish this activity.This is interesting work, with two main contributions: The SPICE pipeline (if made available to the scientific community), and the report of interaction between Ikaros and Jun. However, the distinction between DNA motifs, and the proteins actually binding and having a biological function, should be made clear consistently throughout the manuscript. The same DNA motifs can be bound by multiple factors, for instance within transcription factor families with highly homology in the DNA-binding regions of the proteins.

The reviewer has correctly assessed the content of our manuscript.

Some specific points:SPICE: It is unclear if this is uploaded somewhere to be available to the scientific community.

Thanks for this comment. We will upload the SPICE pipeline and its associated scripts (R and shell) via GitHub.

It was unclear if Ikaros-Jun interaction was initially found from primary Jun ChIP-seq (and secondary Ikaros motif from HOCOMOCO) or from primary Ikaros CHIP-seq (and secondary Jun motif from HOCOMOCO). And - what were the two DNA motifs (primary and secondary, and their distance) from the SPICE analysis?

The IKZF1-JUN interaction was found from primary JUN ChIP-seq data and searching for secondary IKZF1 motifs identified in the HOCOMOCO database. We will provide the primary and secondary motifs in our revised manuscript.

Authors have mostly careful considerations and statements. One additional comment is that binding does not equal function (Fig 2D), and that opening of chromatin (by any other factor(s)) can give DNA-binding factors (like Ikaros and Jun) the opportunity to bind, without functional consequence for the biological process studied.

We appreciate that the reviewer believes our considerations and statement are careful. We agree that opening of chromatin can give the opportunity of factors to bind, and we now make this point in the manuscript.

Figure 2E: Ikaros is reported to be expressed at baseline in murine B cells, yet the Ikaros ChIP-seq in unstimulated cells had what looks to be no significant or low peaks. LPS stimulation induced strong Ikaros ChIP-seq signal. A western blot showing the Ikaros protein levels in the 3 conditions could help understand if the binding pattern is due to protein expression level induction. Similar for Jun (western in the 3 conditions), which seemed to mainly bind in the LPS+IL21 condition. Furthermore, as also suggested below, tracks showing Ikaros and Jun binding from all conditions (unstimulated, LPS only and LPS+IL21 stimulated cells), at select genomic loci, would be helpful in illustrating this difference in signal between the different cell conditions. This is relevant in regards to the point of cooperativity of binding.

The main point of the paper was showing functional cooperation and proximity of binding. However, the use of purified JUN and Ikaros protein suggest cooperative binding. Exhaustive evaluation of the JUN-Ikaros association is left for future studies.

ChIP-seq in mouse B cells showed that Ikaros bound strongly in LPS stimulated cells, in the (relative) absence of Jun binding (Fig. 2C). However, in EMSA (Fig 3C), there is no binding when the AP1 site is mutated, and the authors describe this as Ikaros binding site. What does the Ikaros binding look like at this genomic location in LPS (only) stimulated cells? The authors could show the same figure as in Fig 2F but show Ikaros and Jun ChIP-seq tracks at IL10 CNS9 locus from all conditions to compare binding in unstimulated, LPS and LPS+IL21 cells.

As requested, we now show Ikaros and Jun ChIP-seq tracks from unstimulated, LPS-treated, and LPS + IL21-treated cells. Both Ikaros and cJUN were bound to the Il10 upstream CNS9 region with LPS treatment of cells (see Author response image 1, highlighted in red box), but binding was weaker than that observed with LPS + IL21.

**Author response image 1. sa4fig1:** 

Also: How does this reconcile with the luciferase assay in Fig 4E, where LPS (only) stimulation is used, which in Fig 2E only/mainly induced Ikaros, and not Jun ChIP-seq signal (while EMSA indicate Ikaros cannot bind the site alone, but can enhance Jun-dependent binding).

As shown above, in the LPS (only) condition, both IKZF1 (Ikaros) and cJUN bind to Il10 CNS9 locus. Thus, this is not in conflict with our luciferase assay data in Fig. 4E, which showed Ikaros is dependent on AP-1 binding. Moreover, the AP-1 site in Fig. 4D and 4E can be bound by other AP-1 factors as well, such as JUND, JUNB, BATF, etc. These points can be made in the manuscript. These factors potentially can compete with cJUN binding and their roles remain to be explored.

Comment on statements in results section: The luciferase assays in B and T cells do not demonstrate the role of the proteins Ikaros or Jun directly (page 10, lines 208 and surrounding text). The assay measures an effect of the DNA sequences (implying binding of some transcription factor(s)), but does not identify which protein factors bind there.

We agree with the reviewer. It is reasonable and even likely that different family members may be partially redundant. This point is now made on our revised manuscript.

Lastly, the authors only discuss Ikaros (using the term IKZF1 which is the gene symbol for the Ikaros protein). There are other Ikaros family members that have high homology and that are reported to bind similar DNA sequences (for instance Aiolos and Helios), which are expressed in B-cells and T-cells. A discussion of this is of relevance, as these are different proteins, although belonging to the same family (the Ikaros-family) of transcription factors. For instance, western for Aiolos and Helios will likely detect Aiolos in the B cells used, and Helios in the T cells used.

We agree with the reviewer. As requested, we now discuss the possibility that Aiolos or Helios may also contribute.

**Reviewer #2 (Public Review):**
The study is performed with old tool Spamo (12 year ago), source data from Encode (2010-2012), even peak caller tool version MACS is old ~ 2013. De novo motif search tool is old too (new one STREME is not mentioned). Any composite element search tool published for the recent 12 years are not cited, there are some issues in data analysis in presentation. Almost all references are from about 8-10 year ago (the most recent date is 2019)The title is misleadingInstead of “A new pipeline SPICE identifies novel JUN-IKZF1 composite elements”It should be written as “Application of SpaMo tool identifies novel JUN-IKZF1 composite elements”It reflects the pipeline better but honestly shows that the novelty is missed.

Regarding the above two points, we respectfully disagree with the reviewer. Although SpaMo was used, the pipeline we developed is new and our findings are distinctive. The pipeline can systematically screen and predict novel protein-protein binding complex, and our discovery related to IKZF1-JUN composite element is new and the biological findings and validation are distinctive. This point is now made in the revised manuscript. As requested, we have added some additional references.

The study was performed on too old data from ENCODE, authors mentioned 343 Encode ChIP-Seq libraries, but authors even did not care even about to set for each library the name of target TF (Figure 1E, Figure S2, Table 2).

Although we used ENCODE data, which was in part when we initially developed the algorithm, those data are valid and using them allowed us to demonstrate the functionality of SPICE, which is versatile and can be used on datasets of one’s choice as well. As requested, in the revised manuscript we have added the names of the TFs in Figs, Fig. S2, and Table 1.

**Reviewer #3 (Public Review):**
The authors of this study have designed a novel screening pipeline to detect DNA motif spacing preferences between TF partners using publicly available data. They were able to recapitulate previously known composite elements, such as the AP-1/IRF4 composite elements (AICE) and predict many composite elements that are expected to be very useful to the community of researchers interested in dissecting the regulatory logic of mammalian enhancers and promoters. The authors then focus on a novel, SPICE predicted interaction between JUN and IKZF1, and show that under LPS and IL-21 treatment, JUN and IKZF1 in B cells have significant overlap in their genomic localization. Next, to know whether the two TFs physically interact, a co-immunoprecipitation experiment was performed. While JUN immunoprecipitated with an anti-IKZF1 antibody, curiously IKZF1 did not immunoprecipitate with an anti-JUN antibody. Finally, EMSA and luciferase experiments were performed to show that the two TFs bind cooperatively at an IL20 upstream probe.

The reviewer has described the basic results of the study.

Major strengths:1. SPICE was able to recapitulate previously known composite elements, such as the AP-1/IRF4 composite elements (AICE).1. Under LPS and IL-21 treatment, JUN and IKZF1 in B cells have significant overlap in their genomic localization. This is very good supporting evidence for the efficacy of SPICE in detecting TF partners.

We are glad that the reviewer believes that SPICE is effective in detecting TF partners.

Major weaknesses:1. The authors fail to convincingly show that IKZF1 and Jun physically interact. A quantitative measurement of their interaction strength would have been ideal.

We agree that it is not conclusive that the factors interact directly as opposed to binding to nearby sites on DNA, which is what SPICE was intended to detect. We never intended to claim that we established a definite physical interaction. The coIP worked in one direction, but not reliably in the other, even though we have tried a total of four different antibodies. We now mention in the revised manuscript that we have tried the additional anti-JUN antibodies, cJun (60A8, CST) and JunD (D17G2, CST).

1. The super-shift experiment to show that the proteins bound to their EMSA probe were indeed IKZF1 and JUN are not very convincing and would benefit from efforts to quantify the shift (Figure 3E). Nuclear extracts from cells with single or double CRISPR knock outs of the two TFs would have been ideal.

We agree that using single or double knockouts would be helpful, but other Ikaros family or Jun family members could be involved, so such studies might not be definitive. That is why we used purified proteins to show apparent cooperative binding (Figure 4C).

1. There is a second band beneath the more prominent band in the EMSA experiment with recombinant IKZF1 and JUN (Figure 4C). This second band is most probably bound by IKZF1 because it becomes weaker when the IKZF1 site is mutated and is completely absent when only JUN is added. This is completely ignored by the authors. Therefore, experiments with EMSA fail to convincingly show that IKZF1 and Jun bind cooperatively. They could just as well bind independently to the two sites.

The second band has a faster mobility and might relate to IKZF1, although this is difficult to know. We comment on this band on revised manuscript. As noted above, the purified protein experiments do suggest cooperativity. However, our overall intent was to identify factors binding in proximity, which SPICE has successfully done, even if the binding was “independent”.